# The Role of the Microbiome in First Episode of Psychosis

**DOI:** 10.3390/biomedicines11061770

**Published:** 2023-06-20

**Authors:** Lucero Nuncio-Mora, Nuria Lanzagorta, Humberto Nicolini, Emmanuel Sarmiento, Galo Ortiz, Fernanda Sosa, Alma Delia Genis-Mendoza

**Affiliations:** 1Laboratory of Genomics of Psychiatric and Neurodegenerative Diseases, National Institute of Genomic Medicine, Mexico City 14610, Mexico; lu.nunciom@gmail.com (L.N.-M.); hnicolini@inmegen.gob.mx (H.N.); 2Posgraduate Studies in Biological Sciences, Posgraduate Unit, Posgraduate Circuit, Universitary City, Building D, 1st Floor, Coyoacan, Mexico City 04510, Mexico; 3Carraci Medical Group, Mexico City 03740, Mexico; lanzagorta_nuria@gmc.org.mx (N.L.); fer.sosaher98@gmail.com (F.S.); 4Psychiatric Children’s Hospital Dr. Juan N. Navarro, Mexico City 14080, Mexico; emmanuelsarmientoh@hotmail.com (E.S.); jnnavarrolab@yahoo.com (G.O.)

**Keywords:** microbiome, dysbiosis, first episode of psychosis, brain-gut-microbiome axis

## Abstract

The relationship between the gut-brain-microbiome axis has gained great importance in the study of psychiatric disorders, as it may represent a new target for their treatment. To date, the available literature suggests that the microbiota may influence the pathophysiology of several diseases, including psychosis. The aim of this review is to summarize the clinical and preclinical studies that have evaluated the differences in microbiota as well as the metabolic consequences related to psychosis. Current data suggest that the genera *Lactobacillus* and *Megasphaera* are increased in schizophrenia (SZ), as well as alterations in the glutamate-glutamine-GABA cycle, serum levels of tryptophan, kynurenic acid (KYNA), and short-chain fatty acids (SCFAs). There are still very few studies on early-onset psychosis, thus more studies are needed to be able to propose targeted therapies for a point when the disease has just started or has not yet progressed.

## 1. Introduction

In recent decades, the study of the microbiome has become an important topic in the neuroscientific area because it has been observed that the brain-gut-microbiome axis plays an important role in many psychopathologies such as depression, anxiety, addiction, eating disorders, schizophrenia, autism, and neurodegenerative diseases such as Alzheimer’s disease [1,2].

The microbiota comprises microorganisms (bacteria, viruses, fungi, archaea) that interact with each other within the organism, which has a defined habitat with specific physicochemical properties. The microbiome describes the entire environment, including microorganisms, their genomes, and their metabolites, in a specific ecological niche [3,4].

Species diversity within the gut microbiota can be quantified by sequencing the small subunit ribosomal RNA (16S rRNA) gene in fecal material. This gene is highly conserved and is used as a marker to identify taxa and their relative abundance in a specific niche. Clusters of gene sequences from the microbiota that have more than 97% shared sequence similarity in the 16S rRNA gene are known as operational taxonomic units (OTUs), and the proportion of OTUs that belong to distinct phyla varies across host taxa [5].

The greatest population of microbiota is found in the gastrointestinal tract. The number of bacterial cells in the intestine is estimated to be approximately 3.8 × 10^13^, and it contains 1000–5000 species, most of which are Phyla Firmicutes, Bacteroidetes, Actinobacteria, Proteobacteria, Fusobacteria, and Verrucomicrobia. The gut microbiota has a diversity of genes between 50 and 100 times bigger than the host; the functional expression of these genes is responsible for synthesizing vitamins, enzymatic proteins, etc., which the host does not synthesize. Thus, it works to maintain host homeostasis and facilitates normal metabolism [5,6,7,8,9].

Each microbe has specific functions in the gut, and changes in the community or in the interaction with the host can activate virulence in otherwise commensal microbes, and this could lead to disease. Thus, gain, loss, or changes in the abundance of different taxa of bacteria represent an imbalance that is known as “gut dysbiosis”, which can occur due to disorders related and unrelated to the gastrointestinal tract, use of antibiotics, high-fat and high-sugar diets, and exposure to pesticides and heavy metals [10,11].

The crucial involvement of the microbiome in overall homeostasis makes evident the coevolution of the human host along with such symbionts. Thus, human physiology is the key combination of a host and symbiotic microbiome gene expression, termed the “holobiome” [6].

Due to this close relationship, the microbiome influences the physiological, functional, and behavioral traits of the host [5]. Gut microbial communities have been observed to differ in their metabolic potential from free-living microbial communities. The microbial profile associated with the host varies according to birth, breast-feeding, genetic predisposition, age, diet, environmental factors, stress, infections, and antibiotic use, among others. After the first two years of life, the diversity of the microbiome is more like that of adults. However, it can vary due to the factors mentioned above [12,13,14].

The aim of this review is to summarize the clinical and preclinical studies that have been evaluated to date on the existing differences in the microbiota as well as the metabolic consequences related to psychosis.

## 2. Gut-Microbiota-Brain Axis

It has been observed that there is bidirectional communication between the gastrointestinal tract and the brain in both healthy conditions and diseases. This network is called the gut-microbiota-brain axis. This bidirectional communication is exerted through different communication pathways, such as the vagus nerve, the neuroendocrine system, and the immune system (Figure 1) [15].

The vagus nerve is a paired nerve that consists of sensory (afferent) and motor (efferent) neurons. Within the small and large intestines, vagal afferents innervate the muscle and mucosal layers but do not cross the epithelial layer, so they are not in direct contact with the microbiota in the gut lumen but are in contact with the endocrine cells, which detect the signals through toll-like receptors (TLR4) of bacterial compounds, metabolites, hormones, or neurotransmitters that can be produced by gut microbiota [16,17,18,19].

Likewise, the gut microbiome can not only catalyze the synthesis of neurotransmitters such as dopamine, norepinephrine, γ-aminobutyric acid (GABA), glutamate, and serotonin through dietary metabolism, it can also produce precursors of neurotransmitters (Table 1) [20,21]. However, neurotransmitters such as glutamate, GABA, dopamine, and serotonin do not cross the blood–brain barrier; they must be synthesized in the brain from local pools of neurotransmitter precursors that are derived from the diet (e.g., tyrosine and tryptophan). They enter the blood, are transported across the blood–brain barrier, and are taken up by corresponding neurotransmitter-producing cells. The precursors are then converted into functional neurotransmitters including dopamine, norepinephrine, and serotonin through a number of intermediate steps with the help of various host enzymes. Accordingly, the gut microbiome plays an important role in regulating the metabolism of the neurotransmitter precursors [21].

For instance, *Bacteroides*, *Bifidobacterium*, *Parabacteroides*, and *Escherichia* spp. are known to produce the neurotransmitter GABA [22]. This is important because alterations in central GABA receptor expression are implicated in depression and anxiety. It has been demonstrated that the administration of *Lactobacillus rhamnosus* reduces stress-induced corticosterone levels and the behavior associated with anxiety and depression. Moreover, the neurochemical and behavioral effects are abolished in mice that were vagotomized, suggesting that there is an important communication between bacteria and brain through the vagus nerve [23].

Glutamate is an excitatory neurotransmitter and is synthetized by neuropod cells in the intestine and used to transfer rapid signals to the brain via the vagus nerve [21,24]. The metabolism of glutamate is affected by *Bacteroides vulgatus* and *Campylobacter jejuni*. Furthermore, l-glutamate can be converted to d-glutamate with glutamate racemase and gut bacteria including *Corynebacterium glutamicum*, *Brevibacterium lactofermentum*, and *Brevibacterium avium*, which may affect the glutamate NMDA receptor and cognitive functions in Alzheimer’s disease patients [25].

Dopamine is the most abundant catecholamine neurotransmitter in the brain, and its dysregulation is associated with schizophrenia. Outside the brain, the production of dopamine is mediated by *Staphylococcus* in the human gut. *Staphylococcus* can take up the precursor l-3,4-dihydroxy-phenylalanine (l-DOPA) and convert it into dopamine through staphylococcal aromatic amino acid decarboxylase (SadA) expressed by these bacteria [26,27].

With regard to serotonin, approximately 95% is synthesized in peripheral parts of the human body, mainly by enterochromaffin cells in the intestinal epithelium. Serotonin cannot cross the blood–brain barrier, but its precursor tryptophan can. In the gut, enterochromaffin cells take up tryptophan from dietary protein as a substrate to synthesize serotonin, and this process is regulated by the bacterial kynurenine synthesis pathway [28,29]. Indigenous spore-forming bacteria in the gut (predominantly *Clostridia*) can promote host serotonin biosynthesis and impact gastrointestinal motility and hemostasis [30].

In addition, the gut microbiome participates in the hydrolysis of the different polysaccharides into SCFAs, such as propionate, butyrate, and acetate [31]. The monocolonization of the intestines of germ-free adult mice with *Clostridium tyrobutyricum*, a bacterial strain producing butyrate, or *Bacteroides thetaiotaomicron*, which produces mainly acetate and propionate, decreased blood–brain barrier (BBB) permeability. Moreover, the intravenous or intraperitoneal administration of sodium butyrate has been reported to inhibit histone deacetylation and facilitate long-term memory consolidation, prevent BBB breakdown, and promote angiogenesis and neurogenesis. Whether the gut microbiota and sodium butyrate alter histone acetylation of brain microvascular endothelial cells requires further study to better understand the effects of SCFAs produced by the gut microbiota on the CNS [32].

Microglia, the resident immune cells in the brain, are essential for modulating neurogenesis, influencing synaptic remodeling, and regulating neuroinflammation by surveying the brain microenvironment. In this context, the gut microbiome plays a pivotal role in regulating microglial maturation and function, and altered microbial community composition has been reported in neurological disorders with known microglial involvement in humans [33]. In recent studies, germ-free (GF) mice were colonized with *Bifidobacteroum* spp. and showed that these bacteria can participate in establishing functional neuronal circuits [34].

Furthermore, some aspects of brain development depend on signals from the gut microbiota. For instance, colonization of GF with specific growth phenotype-associated microbial communities can influence early neuron and oligodendrocyte development [35] and alter neurogenesis [36].

On the other hand, the gut contains its own lymphoid tissue called gut-associated lymphoid tissue (GALT), which protects the gastrointestinal tract from invading pathogens. Nonetheless, the immune response to microbiota in the intestine should be tightly regulated to prevent intestinal inflammation. In this context, the intestinal epithelial cell layer acts as a barrier so that the immune cells residing in the intestinal mucosa do not recognize the microbiota present in the intestine [37,38].

GALT is located in the intestinal epithelium, and it is organized in lymphoid follicles in the lamina propria known as Peyer’s patches, where we can also find B cells, T cells, dendritic cells (DCs), and macrophages. The cells that compose the epithelial cell layer (Paneth cells, globet cells, microfold (M) cells) are responsible for the active transport or passive diffusion of antigens from food during digestion and microbial components [38].

Microbial-associated molecular patterns (MAMPs), such as lipopolysaccharide (LPS), bacterial lipoprotein (BLP), and flagellin, among others, can activate DCs, B cells, or T cells. Once activated, they produce inflammatory cytokines (interleukin-1α, interleukin-1β, tumor necrosis factor-alpha (TNFα), and interleukin-6) that cross the BBB via both diffusion and cytokine transporters (Figure 1). In the brain, these cytokines act on receptors expressed in neurons and glial cells, specifically in microglia, altering their state of activation and function. In turn, cytokines can act on the receptors of different afferent nerves, promoting alterations in signals, from the digestive tract to the CNS [39].

Although some bacteria can cause these inflammatory responses, others, such as segmented filamentous bacteria, are necessary for the induction of intestinal T-cell expansion and maturation of the immune system. For example, *Clostridia* are necessary for the production of Treg cells, which induce anti-inflammatory molecules such as interleukin-10 (IL-10) that help maintain the homeostasis of the gut [40,41].

Finally, chronic stress plays an important role in the hypothalamic-pituitary-adrenal (HPA) axis by releasing corticotropin-releasing hormone (CRH) from the hypothalamus. This information is then processed by the anterior lobe of the pituitary gland, resulting in the secretion of adrenocorticotropic hormone (ACTH). This stimulates the release of cortisol into the blood from the adrenal cortex. An increase in the concentration of cortisol in the blood has a beneficial effect on the body, but a prolonged increase in its concentration is harmful and could affect the integrity of the intestinal barrier, which can alter the gut environment and change the composition of the microbiota, leading to dysbiosis [18,42].

A healthy and stable gut microbiota community plays a vital role in maintaining the homeostatic balance in the gut microbiota-brain axis, as well as gut barrier integrity, immunity, and metabolism [43,44,45]. Together, this intestinal eubiotic state is characterized by a preponderance of potentially beneficial species, belonging mainly to the two bacterial phyla Firmicutes and Bacteroides, while potentially pathogenic species, such as those belonging to the phylum Proteobacteria (Enterobacteriaceae), are present, but in a very low percentage very [46].

As mentioned above, when dysbiosis occurs, the “good bacteria” no longer regulate the “bad bacteria,” and the latter take over, causing a change in the intestinal metabolite profile that has been linked to developing psychotic disorders [46,47].

## 3. Gut Microbiome and Psychosis

The gut microbiome has been shown to be involved in synthesizing neurotransmitters like GABA, dopamine, glutamate, and serotonin, which have been implicated in mental disorders like depression, schizophrenia, and bipolar disorder [48,49,50].

Psychosis is a common characteristic of a broad range of psychiatric disorders. It is a feature of schizophrenia spectrum disorders, as well as neurodegenerative conditions and mood disorders, although it can also develop as a result of intoxication, seizure disorders, central nervous system infections, autoimmune disorders, genetic syndromes, medication side effects, neoplasties, and substance abuse. Other than substance abuse, most medical conditions associated with psychosis are rare [51,52,53].

According to the American Psychiatric Association (APA) and the World Health Organization (WHO), a diagnosis of psychosis requires disorganized speech, catatonic behavior, and impaired reality, i.e., the presence of hallucinations (perceptions that occur in the absence of corresponding somatic or external stimuli) and delusions (persistent false beliefs) [54,55].

Onset is usually between the ages of 18 and 20 years of age. However, some individuals can present with it before the age of 18 (first episode of psychosis), and it is usually associated with chronic morbidity and functional impairment [53].

About 8% of teenagers (13 to 18 years) and 17% of children (9 to 12 years) describe experiences similar to psychosis [56]. However, its evaluation in youth presents many challenges, one of which is that hyperactive imagination and vivid fantasies can be misinterpreted as psychosis. Therefore, the age, culture, and cognitive development of the adolescent must be considered when evaluating reports of suspected psychosis [53].

However, its evaluation in childhood and adolescence is important to minimize the long-term impact of psychiatric disorders. For this reason, efforts are being made to develop biomarkers based on the observation of changes in the composition of the microbiome for use in psychiatry in patients of these ages, since it has clinical appeal, due to sample collection being non-invasive [57]. Relatedly, studies are using experimental approaches in animal models to study and manipulate the gut microbiome to improve psychiatric health outcomes, including fecal microbial transplantation into GF mice with human or specific microbiota and gut microbiome modification with antibiotics and probiotics [1].

Table 2 summarizes the most important studies that were present in this review and the metabolic implications that were found in them.

### 3.1. Schizophrenia

In this context, studies were conducted using mice colonized with the microbiota of patients with schizophrenia (SZ) to investigate whether the behavioral phenotype of the disease is related to changes in the microbiota, and psychomotor behavioral abnormalities such as hyperactivity and changes in learning and memory were observed.

Specifically, transplantation of *Streptococcus vestibularis*, a bacterium that is elevated in patients with schizophrenia, resulted in deficits in social behavior, altered glutamate-glutamine-GABA cycles, and tryptophan-kynurenine metabolism in recipient mice. This suggests that changes in the gut microbiota are associated with some, but not all, of the endophenotypes characteristic of mouse models of SZ [58,59].

In the case of the probiotic evaluation, a 14-week probiotic treatment trial was conducted in patients with schizophrenia with their usual antipsychotic treatment, where the probiotic mixture included *Lactobacillus rhamnosus* and *Bifidobacterium animalis.* Supplementation led to an improvement in intestinal epithelial integrity [60].

Similarly, a combination of vitamin D and a probiotic mixture containing *Bifidobacterium bifidum*, *Lactobacillus acidophilus*, *Lactobacillus fermentum*, and *Lactobacillus reuteri* given to patients with SZ for 12 weeks resulted in significant reductions in C-reactive protein, metabolic changes, and improvements in positive and negative symptom scales [61].

Related to prebiotic treatment, a study was conducted in rats evaluating the prebiotic galactooligosaccharide bimunoTM (B-GOS^®^) as an adjunct to olanzapine treatment, which is commonly used in patients with schizophrenia. Its use in combination with olanzapine reduces acetate concentrations, prevents weight gain, and has benefits on cognitive function in psychosis [62]. However, the study of fecal microbiome transplantation, as well as the use of probiotics and prebiotics, still needs to be explored [62].

The number of previous investigations of the gut microbiome in people with schizophrenia is limited, and most have been performed in adults. To date, studies examining differences between healthy controls and individuals with schizophrenia have reported significant differences in the abundance of certain taxa [63]. For example, in China, three investigations found that in patients with schizophrenia, the genera *Succinivibrio*, *Megasphaera*, *Clostridium*, and *Collinsella* are increased with respect to controls. They further observed that *Succinivibrio* is related to symptom severity [64,65,66]. Nguyen et al. [67] observed that at the genus level, *Anaerococcus* relatively increased in schizophrenia, unlike *Haemophilus*, *Sutterella*, and *Clostridium* which decreased. At the same time, within individuals with schizophrenia, *Ruminococcaceae* abundance correlated with lower severity of negative symptoms [67].

In another study in the same country, *Veillonellaceae*, *Prevotellaceae*, *Bacteroidaceae*, and *Coriobacteriaceae* families increased, compared to *Lachnospiraceae*, *Ruminococcaceae*, *Norank*, and *Enterobacteriaceae*, which decreased in schizophrenia patients [59].

On the other hand, He et al. [68] investigated subjects at high and ultra-high risk of schizophrenia as well as healthy controls, finding that the levels of *Clostridiales*, *Prevotella*, and *Lactobacillus ruminis* in the gut microbiota were higher in subjects at ultra-high risk of schizophrenia than in the other two groups. *Clostridiales* and *Prevotella* are involved in carbohydrate fermentation and are important producers of SCFAs, while *Lactobacillus ruminis* stimulates TNFα production. TNFα and SFCAs can cross the BBB and activate microglia, and this in turn induces excessive synaptic pruning, apoptosis, BBB disruption, and neuroinflammation, which has been found to be associated with SZ [68].

Zhu et al. [58] also examined the fecal microbiota of patients with first-episode schizophrenia without medication and found an enrichment of 11 bacterial species: *Akkermansia muciniphila*, *Bacteroides plebeius*, *Veillonella parvula*, *Clostridium symbiosum*, *Eubacterium siraeum*, *Cronobacter sakazakii/turicensis*, *Streptococcus vestibularis*, *Alkaliphilus oremlandii*, *Enterococcus faecium*, *Bifidobacterium longum*, and *Bifidobacterium adolescence* [58].

In this study, they also performed an analysis of patients after 3 months of treatment with risperidone or other antipsychotics and found that 12 operational taxonomic units (OTUs) remained significantly altered compared to controls, i.e., that antipsychotic treatment affects the gut microbiota but does not completely restore the altered microbiota associated with SZ [58].

Particularly, a review by Liu et al. [1] observed that the genera that were increased compared to controls were *Fusobacterium*, *Lactobacillus*, *Megasphaera*, and *Prevotella*, most of which are Gram-negative bacteria Figure 2 [1]. Although Gram-negative bacteria are common in the gut microbiome, their increase can lead to the systemic circulation of enteric inflammatory molecules, such as lipopolysaccharides, which are an effective neurodevelopmental model of schizophrenia in rodents [69,70,71]. This causes an increase in inflammatory cytokines, which may contribute to the alteration in intestinal permeability [72].

### 3.2. First Psychotic Episode (FEP)

With regard to patients with a first psychotic episode (FEP), approximately 25 taxonomic differences between them and control patients have been identified. Among those that stand out is the increase of *Protobacteria* at the genus level, Lactobacillaceae at the family level, and *Clostridium coccoides* at the species level in FEP patients [73,74,75].

A study conducted in Finland assessed the microbiota of patients with psychosis treated with olanzapine, risperidone, or quetiapine and found a significant increase in the Lactobacillaceae, Halothiobacillaceae, Brucellaceae, and Micrococcineae families and a decrease in Veillonellaceae compared to controls [73].

Several studies have also been conducted in China to determine whether antipsychotic treatment affects the composition of the gut microbiota and to compare changes in taxa in patients with psychosis and controls. For example, Ma et al. [76] compared the microbiota of patients with an untreated first psychotic episode, patients chronically treated with antipsychotics, and controls and found that patients with psychosis, both treated and untreated, had distinct changes in certain taxa, such as an increase in Christensenellaceae, Enterobacteriaceae and Victivallaceae, while Pasteurellaceae, Turicibacteraceae, Peptostreptococcaceae, Veillonellaceae, and Succinivibrionaceae were decreased compared to controls. In addition, they observed that patients chronically treated with antipsychotics had a higher abundance of the families Peptostreptococcaceae and Veillonellaceae and the genera *Megasphaera*, *Fusobacterium*, and *SMB53* compared to patients with a first psychotic episode who were not yet treated [76].

Another study in the same country compared fecal bacteria in patients with a first psychotic episode before and after 24 weeks of risperidone treatment and found that patients had decreased levels of *Bifidobacterium* spp., *Escherichia coli*, and *Lactobacillus*, while *Clostridium coccoides* was increased compared to controls [74].

Studies have also been conducted in patients not treated with antipsychotics, which were characterized by an increased abundance of harmful bacteria (Proteobacteria) and a decrease in short-chain fatty acid-producing bacteria, such as the genera *Faecalibacterium* and *Lachnospiraceae.* It has been observed that short-chain fatty acids are important for maintaining the homeostasis of the intestinal barrier, so if they are not produced, this could lead to the translocation of opportunistic gut pathogens (mainly Proteobacteria) to the mesenteric lymph nodes or the bloodstream, triggering an inflammatory response [75]. In addition, the low presence of *Faecalibacterium* in the gut leads to an increase in intestinal Th17 cells, which produce pro-inflammatory cytokines that can cross the blood-brain barrier and migrate to the brain, where they activate hippocampal microglia and induce abnormal behavior [75,77].

**Table 2 biomedicines-11-01770-t002:** The most important studies that are presented in this review and the metabolic implications that have been found in them.

Reference	Changes in Microbiota	Type of Study	Metabolic Effect
Schizophrenia			
Ghaderi et al., 2019 [61]	Not reported	Human (SZ in treatment with Vitamin D and probiotics)	Decrease in fasting plasma glucose, insulin, triglycerides, VLDL-total, LDL-total, HDL-cholesterol
He et al., 2018 [68]	No differences in α diversity but differences in β diversity. in both SZ patients Genus enriched in UHRs: *Lactobacillus* and *Prevotella* Species enriched in UHR: *Lactobacillus ruminis*	Human (High-risk schizophrenia patients (HR) vs. ultra-high risk- schizophrenia (UHR) vs. controls)	The initiation pathways of pyruvate synthesis, acetyl-CoA synthesis, and fatty acid biosynthesis were increased in the UHR.
Kao et al., 2018 [62]	Use of B-GOS^®^ alone increased *Bifidobacteria* spp. and decreased *Escherichia/Shigella* spp., *Coprococcus* spp., *Oscillibacter* spp., *Coccoides* spp., *Roseuria Intestinalis Cluster*, and *clostridium XVIII cluster.* No changes in microbiota with intake olanzapine alone.	Animal (rats)	BGOS^®^ and olanzapine alone increased plasma acetate concentrations. Combined administration of BGOS and olanzapine decreased plasma acetate concentrations.
Li et al., 2020 [64]	No differences in α diversity but differences in β diversity. Genus enriched in SZ: *Collinsella*, *Lactobacillus*, *Succinivibrio*, *Mogibacterium*, *Corynebacterium*, undefined *Ruminococcus* and undefined *Eubacterium.*	Human (SZ in treatment with antipsychotics vs. controls)	In the SZ group, the pathways ascorbate and aldarate metabolism, nucleotide metabolism, and propanoate metabolism were enriched.
Nguyen et al., 2019 [67]	No differences in α diversity but differences in β diversity. Genus enriched in SZ: *Anaerococcus*	Human (SZ in treatment with antipsychotics vs. controls)	Not reported
Shen et al., 2018 [65]	No differences in α diversity but differences in β diversity. Genus enriched in SZ: *Succinivibrio*, *Megasphaera*, *Collinsella*, *Clostridium*, *Klebsiella*, and *Methanobrevibacter.*	Human (SZ in treatment with antipsychotics vs. controls)	Pathways associated with *Clostridium*: beta alanine metabolism, butanoate, phenylalanine, and inorganic ion transport. Pathways associated with *Collinsella*: tyrosine and selenocompound metabolism.
Xu et al., 2020 [66]	Lower α diversity in SZ. Genus enriched in SZ: *Eggerthella* and *Megasphaera*, Species enriched in SZ: *Akkermansia muciniphila*, *Bifidobacterium adolescentis*, *Clostridium perfringens*, *Lactobacillus gasseri*, and *Megasphaera elsdeniis*.	Human (SZ vs. controls)	Activity of gut glutamate synthetase elevated in SZ patients.
Zheng et al., 2019 [59]	Lower α diversity in SZ. Families increased in patients with SCZ: Veillonellaceae, Prevotellaceae, Bacteroidaceae, and Coriobacteriaceae.	Human (SZ vs. controls) and Animal (mice transplanted with microbiota of SZ patients)	Mice with fecal transplantation for the SCZ have alterations in the glutamate-glutamine-GABA cycle and amino acid metabolism and transport. In addition, lipids were decreased in the serum and hippocampus of transplanted mice.
Zhu et al., 2020 [58]	Increased in β and α diversity in SZ. Species enriched in SZ without medication: *Akkermansia muciniphila*, *Bacteroides plebeius*, *Veillonella parvula*, *Clostridium symbiosum*, *Eubacterium siraeum*, *Cronobacter sakazakii/turicensis*, *Streptococcus vestibularis*, *Alkaliphilus oremlandii*, *Enterococcus faecium*, *Bifidobacterium longum*, and *Bifidobacterium adolescence*.	Human (First episode of schizophrenia vs. 3 months later with antipsychotics vs. controls) and Animal (mice)	Transplantation of *Streptococcus vestibularis* resulted in altered glutamate-glutamine-GABA cycles and tryptophan-kynurenine metabolism.
First Episode of Psychosis (FEP)			
Ma et al., 2020 [76]	Low α diversity in SZ treated with antipsychotics. Families increased in FEP and SZ with treatment: *Christensenellaceae*, *Enterobacteriaceae*. Genus enriched in FEP and SZ with treatmente: *Escherichia*.	Human (FEP drug naïve vs. SZ treated with antipsychotics vs. controls)	Not reported
Schwarz et al., 2018 [73]	Genus increased in FEP patients: *Lactobacillus*, *Tropheryma*, *Halothiobacillus*, *Saccharophagus*, *Ochrobactrum*, *Deferribacter*, and *Halorubrum*.	Human (FEP in treatment with antipsychotics vs. controls)	Not reported
Yuan et al., 2018 [74]	*Bifidobacterium* spp. and *Escherichia coli* increased with risperidone treatment. Species increased in FEP: *Clostridium coccoides.*	Human (FEP drug naïve vs. FEP in treatment with risperidone vs. controls)	After 24 weeks of treatment with risperidone there were increases in weight, fasting serum levels of glucose, triglycerides, LDL, and protein C-reactive.
Zhang et al., 2020 [75]	No differences in α diversity but differences in β diversity. Increase proteobacteria in FEP drug naïve.	Human (FEP drug naïve vs. controls)	Not reported

## 4. Discussion

It has become increasingly clear that the microbiome affects human health and is considered an important factor in influencing neuroendocrinological function and brain development, besides being involved in the pathophysiology of some mental disorders, such as anxiety, depression, and schizophrenia [63,78].

This review is based on clinical and preclinical studies in SZ and FEP. As can be seen in Table 2, most of the studies presented did not show changes in alpha diversity in patients with schizophrenia or FEP compared to controls. However, there are significant differences in beta diversity, which may indicate a different composition of bacteria involved in SZ and FEP.

As mentioned above, the most common genera in the reviewed studies were *Lactobacillus* and *Megasphaera*. *Megasphaera* is associated with cognitive impairment and inflammation in patients with hepatic encephalopathy, while increased *Lactobacillus* is associated with TNF production and intestinal inflammation [66,68].

In terms of pathophysiological changes, alterations in the glutamate-glutamine-GABA cycle and increased SCFAs have been observed in patients with schizophrenia (Table 2). Glutamatergic neurotransmission and hippocampal glutamate reductions are involved in SCZ [59]. SCFAs can cross the BBB and activate (propionate) or inhibit microglia in addition to favoring the inflammatory process. This is related to the membrane hypothesis, where it is suggested that activation of microglia induces excessive synaptic pruning, apoptosis, blood–brain barrier changes, and neuroinflammation [68].

Hence, changes in the gut microbiome in FEP or SZ may underlie microglial activation and subsequent altered membrane metabolism in the brain [68]. Furthermore, altered microbiota in psychosis induces lower serum tryptophan, higher serum kynurenic acid (KYNA) levels, and changes in gray matter volume in non-medicated psychotic patients [58].

Therefore, it is important to continue to study the composition of the gut microbiome in children or adolescents with early-onset psychosis to provide further promise for symptom and functional recovery. It is also important to try to avoid these inflammatory imbalances that can cause changes in the BBB and thus trigger a cytokine response that acts on microglia, altering their activation and function [39,68].

However, it should be noted that the composition of the gut microbiome in patients with psychosis shows abnormalities that may be influenced by various demographic, environmental, lifestyle, and treatment factors [79]. As is the case with the dietary patterns of any population, these are influenced by cultural beliefs, available local resources, and agricultural practices that result in specific dietary patterns [80]. Therefore, homogeneity of the gut microbiota may not be observed when different populations resettle in geographically distinct regions. These and other factors, such as sanitation and healthcare access, often drive microbiome differences [81].

Furthermore, most of the papers included in this review are of Asian origin, which is important to note, as understanding these diseases requires the assessment of genetic diversity that interacts with the microbiome [82]. This is demonstrated in a study by Blekhman et al. [83], which examined the genetic and microbial composition of nearly 100 subjects and found a positive correlation between genome sequence similarity and microbiome [83].

Regarding genetic studies, many have focused on participants of Northern European descent. However, each population is different; for example, Hispanic and Latino individuals comprise a broad ethnic group with varying proportions of a mixture of Native American, African, and European ancestry, which may affect susceptibility to metabolic and psychiatric disorders. For this reason, it is necessary to study the microbiome of each population, as some of the inconsistencies may be due to the ethnic ancestry present in the populations [84].

Although gut microbiota composition shows great diversity and interindividual variation, many microbial metabolic gene expression patterns are quite similar [85]. However, as mentioned above, it is necessary to evaluate the microbiome of each specific population in order to identify the taxa involved in early-onset psychosis and thus be able to establish possible causal relationships as well as more targeted treatment [79].

In a different area, fecal microbiota transplantation (FMT) is a promising therapy in which the complete fecal microbiota of a healthy donor, in the form of a solution, is administered into the intestines of the recipient by enema in order to reverse the disequilibrium of the intestinal microbiota of the recipient. FMT has been designated as an Investigational New Drug by the U.S. Food and Drug Administration (FDA). The process involves screening for bacterial, viral, or protozoal infections and the initial evaluation of a healthy donor with no family history of metabolic, autoimmune, or cancer diseases [86]. Although FMT has shown promise in psychiatric disorders and has been studied in murine models,, further research is needed to understand the underlying mechanisms as well as to characterize and account for ethnic variations that exist among populations to safely apply FMT [87].

## 5. Conclusions

This review analyzes the role of the microbiome in the gut-brain axis and provides insight into how it can trigger mental disorders from the gut. Recent studies of the microbiota have opened new possibilities for the treatment of psychiatric disorders. Although there are still inconsistencies regarding the bacteria involved in psychosis, there is evidence that there are differences between patients with psychosis and controls. Therefore, it is important to continue research to develop targeted therapies (such as FMT, probiotics, and prebiotics) for patients with early-onset psychosis, when the disease has just begun or has not yet progressed.

Finally, it is necessary to point out that one of the complications of these studies is the small population sample obtained. It is challenging to obtain samples or recruit patients, hence the need to raise awareness about these new therapies and the potential benefits they may provide.

## Figures and Tables

**Figure 1 biomedicines-11-01770-f001:**
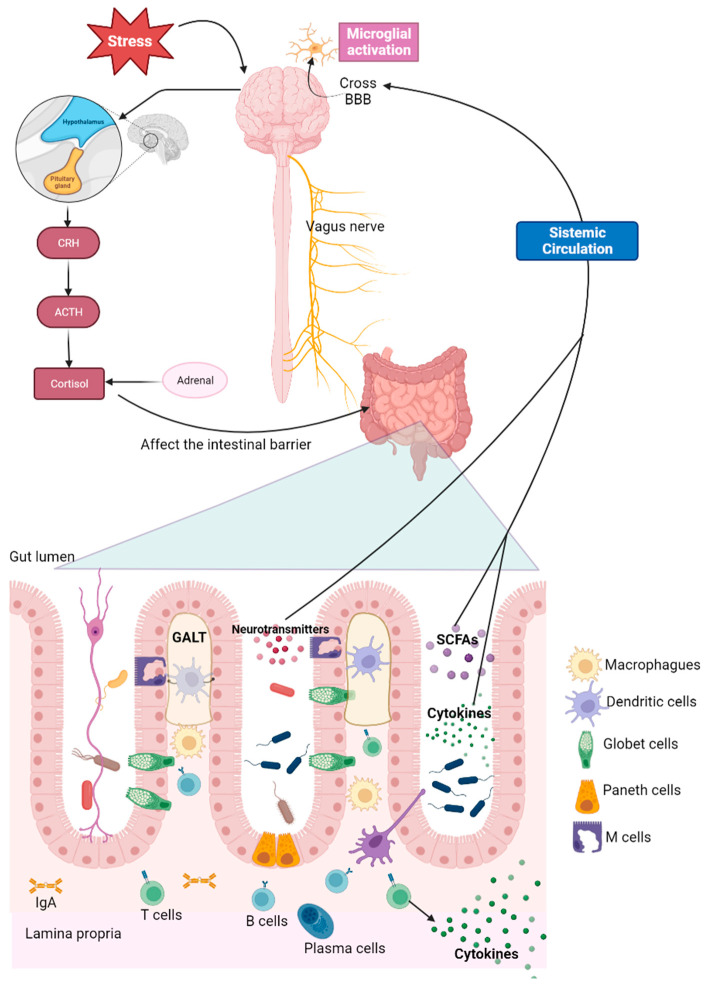
Brain-gut microbiome axis. Bidirectional communication of the microbiome with the brain through the immune system, vagus nerve, and neuroendocrine system. Corticotropin-releasing hormone (CRH), adrenocorticotrophic hormone (ACTH), gut-associated lymphoid tissue (GALT), short-chain fatty acid (SCFAs), immunoglobulin A (IgA), blood–brain barrier (BBB). Created with BioRender.com.

**Figure 2 biomedicines-11-01770-f002:**
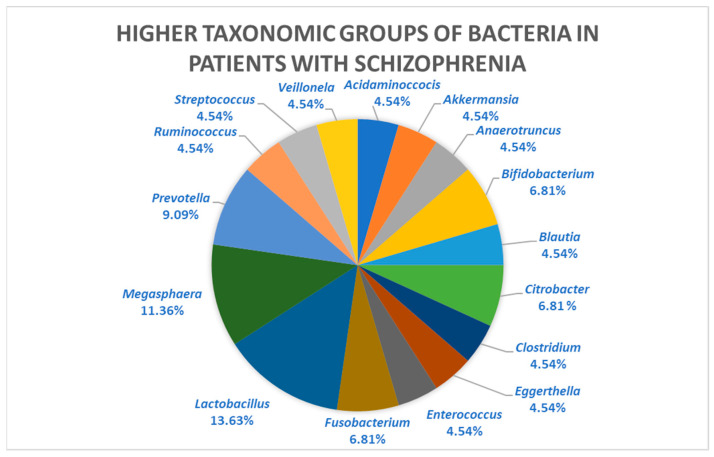
Taxonomic groups of bacteria were found to be elevated in adult patients with schizophrenia relative to controls. It is observed that the genus present in the highest proportion is *Lactobacillus* 13.63%, followed by *Megasphera* 11.36%, *Prevotella* 9.09%, and *Fusobacterium* 6.81%. Adapted from Liu et al., 2021 [1].

**Table 1 biomedicines-11-01770-t001:** Microbiota-derived molecules and its repercussion in the nervous system.

Gut Microbiota	Microbiota-Derived Molecules	Nervous System and Behavioral Changes
*Bacteroides*, *Bifidobacterium*, *Parabacteroides*, and *Escherichia* spp.	GABA *	Depression and anxiety
*Bacteroides vulgatus*, *Campylobacter jejuni*, *Corynebacterium glutamicum*, *Brevibacterium lactofermentum*, and *Brevibacterium avium*	Glutamate	Alzheimer’s disease
*Staphylococcus*	Dopamine	Schizophrenia
*Clostridium tyrobutyricum*, *Bacteroides thetaiotaomicron*	SCFAs *	Decreased blood brain barrier permeability, promote angiogenesis and neurogenesis

* γ-aminobutyric acid (GABA), short-chain fatty acid (SCFAs).

## Data Availability

Not applicable.

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
