# Peer review of "The Role of the Microbiome in First Episode of Psychosis"

_biomedicines, 2023, doi:10.3390/biomedicines11061770_

Round 1

Reviewer 1 Report

Review of

The role of microbiome in first episode of psychosis

              This paper addresses a topic of increasing importance in an evolving literature which remains limited.  The review covers important new research in a concise and well-focused manner.  The writing needs to be improved, both in terms of word choice and sentence structure (many phrases are punctuated incorrectly as complete sentences.

Author Response

Dear Reviewer:
First of all, thank you for your comments, we have rephrased and checked the structure of the sentences.

Reviewer 2 Report

This is a well written and informative review; while it doesn't necessarily move the field forward, it can be a useful addition to the literature, particularly for those not familiar with the field. 

One clarification that will be useful is that the authors are choosing to examine psychosis as a primary outcome- it is important to clarify that  these species may also be related to other psychiatric disorders either independently or through comorbidity with psychosis. 

Author Response

Dear Reviewer:
First of all, thank you for your comments, as you well mention that this microbiota is related to other psychiatric diseases. We have added it in line 211.

Reviewer 3 Report

The Manuscript: „The role of microbiome in first episode of psychosis’’ by Lucero Nuncio-Mora and colleagues attempted to identify the relationship between the microbiome and the first episode of psychosis based on the review of previously published literatures. In addition, the authors summarize the differences between the taxonomic groups of the intestinal microbiome among patients with psychosis and healthy subjects. It has been already established that microbes are responsible for triggering cognitive and emotional activities through the microbiota–gut–brain axis.  The association between alterations in the gut microbiome and several psychiatric conditions, such as autism, depression, bipolar disorder and psychosis has also been repeatedly reported. The authors have analysed these aspects in detail in the submitted manuscript. After going through the manuscript, I have a few comments for the authors:

1.     The Abstract is very vague and short. It does not provide a brief overview of the manuscript. I would suggest the authors to rewrite the abstract with ample overview of the methodology, results and conclusion of the study.

2.     Please mention the aim of the study in the closing sentence of the introduction section.

3.     A brief discussion on disease-specificity and potential causal relationships between changes of the microbiome and disease pathophysiology is missing. Please include this aspect in the discussion section.

4.     There are few literatures addressing diet-microbiota-host interactions and drug-microbiota-host interactions in psychosis. Please discuss these aspects in the manuscript

Author Response

Dear Reviewer:

First of all, thank you for your comments

  1. We rewrite the abstract with more detail in the aim of the study, results and conclusions
  2. We add the aim of the study at the end of the introduction
  3. We have added the relationship between microbiota and disease pathophysiology to the discussion in line 410.
  4. We add and detail from line 268 the studies that have been conducted to date on diet and antipsychotics that interfere with the microbiota of patients with schizophrenia and FEP.

Reviewer 4 Report

The authors present a review on the gut microbiome and psychosis. Overall the review seems to be lacking depth and detail for a review that would assure a reader that this is a high-level, quality review of the field in question.

-As a review the article lightly touches on the microbiome while an in-depth description may be useful for those interested in using your article to increase their background knowledge of the microbiome and apply it to psychosis. This would be important, for example, for psychiatrists that want to use your article to better understand the microbiome.

-to that end, figure 1 should include all of the mechanisms you discuss in the associated paragraph such as NTs and SCFAs, etc.

-can you provide background or discussion on the medication used in psychosis and any studies of their effects on the microbiome? This is especially relevant since you do have a paragraph on FEP which is the classic model for pre-medication investigations.

-Although your review article is positioned on microbiome and psychosis, the review of such studies is sparse and not-detailed. Was this an exhaustive presentation of studies in the area? Are you focusing on both pre-clinical and clinical studies? I would at least expect a table or some figure summarizing findings across many studies. I cannot tell if you just selected a few studies to highlight or are presented the field of evidence to date.

-figure 2 is nice, but it is just one study correct? Consider citing this study in figure 2's footnote.

-for the statement "Additionally, most of the samples included as part of this analysis are of Asian descent, which is important to emphasize, since to understand these diseases, it is necessary to assess the genetic diversity that interacts with the microbiome" - What are you referring to here? What analysis? which study did you base your analysis on?

Author Response

Dear Reviewer:
First of all, thank you for your comments:

-We took a deeper and more comprehensive look at the studies that have been done to date linking microbiota to psychosis.

-We modify Figure 1 with respect to the explanation of the mechanisms mentioned in the text.

-We add literature and explain in more detail how antipsychotics affect the microbiota of patients

-We have compiled the literature to date on preclinical and clinical studies of microbiota and patients with psychosis, with a particular focus on clinical studies, because it is intended that this research can be applied in the future, for example to treatment with probiotics. In line 388 we have added a table of the studies that have been done so far on the microbiota in patients with SZ and FEP.

-Figure 2 is based on a review article that summarizes in a table the taxa found to be elevated in patients with schizophrenia. We have added the citation at the end of Figure 2.

-Our point here is that most of the papers in this review evaluating psychosis and the microbiota to date are of Asian origin. To make it clearer, we rewrite the sentence

Round 2

Reviewer 4 Report

Thank you for making revisions to your manuscript based on reviewer comments. The new version reads well and is much more detailed in its review of the topic.

Author Response

Dear Reviewer:

Thank you for your comments.

Best regards.